# A Quantitative ^1^H NMR Method for Screening Cannabinoids in CBD Oils

**DOI:** 10.3390/toxics9060136

**Published:** 2021-06-10

**Authors:** Ines Barthlott, Andreas Scharinger, Patricia Golombek, Thomas Kuballa, Dirk W. Lachenmeier

**Affiliations:** Chemisches und Veterinäruntersuchungsamt (CVUA) Karlsruhe, Weißenburger Straße 3, 76187 Karlsruhe, Germany; ines.barthlott@student.kit.edu (I.B.); andreas.scharinger@cvuaka.bwl.de (A.S.); patricia.golombek@cvuaka.bwl.de (P.G.); thomas.kuballa@cvuaka.bwl.de (T.K.)

**Keywords:** cannabidiol (CBD), ∆^9^-tetrahydrocannabinol (∆^9^-THC), cannabinol (CBN), ∆^8^-tetrahydrocannabinol (∆^8^-THC), cannabinoids, CBD oil, nuclear magnetic resonance spectroscopy (NMR), PULCON methodology, ^1^H NMR, qNMR

## Abstract

Toxicologically relevant levels of the psychoactive ∆^9^-tetrahydocannabinol (∆^9^-THC) as well as high levels of non-psychoactive cannabinoids potentially occur in CBD (cannabidiol) oils. For consumer protection in the fast-growing CBD oil market, facile and rapid quantitative methods to determine the cannabinoid content are crucial. However, the current standard method, i.e., liquid chromatography combined with tandem mass spectrometry (HPLC-MS/MS), requires a time-consuming multistep sample preparation. In this study, a quantitative nuclear magnetic resonance spectroscopy (qNMR) method for screening cannabinoids in CBD oils was developed. Contrary to the HPLC-MS/MS method, this qNMR features a simple sample preparation, i.e., only diluting the CBD oil in deuterochloroform. Pulse length-based concentration determination (PULCON) enables a direct quantification using an external standard. The signal intensities of the cannabinoids were enhanced during the NMR spectra acquisition by means of multiple suppression of the triglycerides which are a major component of the CBD oil matrix. The validation confirmed linearity for CBD, cannabinol (CBN), ∆^9^-THC and ∆^8^-THC in hemp seed oil with sufficient recoveries and precision for screening. Comparing the qNMR results to HPLC-MS/MS data for 46 commercial CBD oils verified the qNMR accuracy for ∆^9^-THC and CBD, but with higher limits of detection. The developed qNMR method paves the way for increasing the sample throughput as a complementary screening before HPLC-MS/MS.

## 1. Introduction

In recent years, hemp products have experienced a strong increase in popularity. Besides hemp-based foods (e.g., hemp seed oil, hemp flour), consumer products containing cannabidiol (CBD) and in particular “CBD oils” are currently high in demand [1] and the market for CBD-containing dietary supplements continues to grow [2]. The term “CBD oil” originates from the non-psychoactive cannabidiol (CBD) which naturally occurs in the hemp plant. CBD is researched for a variety of pharmacological effects and is approved as medicinal product for treatment of certain epileptic conditions [3]. CBD is also marketed using unauthorized health or disease-related claims [1,2]. Despite the lack of clinical evidence, CBD is advertised as a natural remedy for treating anxiety, depression, pain, inflammatory and sleep disorders, and even cancer [2,4]. CBD oils are formulated as mixtures of an edible oil with extracts of the leaves and flowers of the hemp plant *Cannabis sativa* L. Commercially available CBD oils usually declare a CBD content between 5 and 20 wt.-% [4]. Oils with CBD content at this level can only be achieved by adding a concentrated hemp extract to an edible oil, e.g., hemp seed oil or olive oil [4]. As the extraction process for manufacturing so-called full spectrum hemp extracts is not CBD-selective [4,5], CBD oils contain a wide spectrum of cannabinoids naturally occurring in the hemp plant.

The substance class of cannabinoids refers to terpeno-phenolic C_21_ and C_22_ compounds that are exclusively found in the hemp plant. They are predominantly formed in the glandular hairs of the female hemp plant [6,7]. The occurrence of a carboxyl group allows a further division into two subclasses: cannabinoid acids featuring a carboxyl group (e.g., ∆^9^-tetrahydocannabinolic acid (THCA) and cannabidiolic acid (CBDA)) and the neutral cannabinoids. More than 120 different cannabinoids and their carboxylic acid analogs and conversion products are described in the literature [8]. The individual cannabinoids differ only slightly in their structures. The modifications are mainly limited to changes of the allylic C_5_ side chain (e.g., ∆^9^-tetrahydrocannabivarin (THCV), in Figure 1), the substitution of a carboxylic acid or hydroxyl group, or an additional cyclization [9].

In the hemp plant, CBD and ∆^9^-THC are the most abundant cannabinoids beside cannabigerol (CBG) and cannabinol (CBN), see Figure 1. While CBD and CBG are not psychoactive, ∆^9^-THC induces intense states of intoxication [8]. Consequently, the ∆^9^-THC content in CBD oil must be kept low even at high CBD concentrations for consumer safety. Since the hemp extraction is not typically cannabinoid-selective, the naturally occurring content is the major factor for ∆^9^-THC contents.

In the European Union (EU), fiber hemp with low ∆^9^-THC content (characterized by ratio of (THC + CBN)/CBD < 1) is used for the production of hemp extracts [4] as opposed to drug hemp (ratio of (THC + CBN)/CBD > 1) [1,8,10]. However, Lachenmeier et al. (2020) recently showed that some CBD products sold in Germany, including CBD oils, exceeded the acute reference dose (ARfD) of 1 µg THC/kg body weight established by the European Food Safety Authority (EFSA) [11]. Consequently, these products were classified in part as being not safe for human consumption [12]. Up to 30 mg ∆^9^-THC/daily dose was determined in the products [12]. As CBD oils may contain very high amounts of non-psychoactive cannabinoids and of the psychotropic cannabinoid ∆^9^-THC at toxicologically relevant levels higher than the ARfD, suitable methods are required for the quantitative determination of these cannabinoids.

For the quantification of ∆^9^-THC and other cannabinoids in hemp products, gas chromatography (GC) or high performance liquid chromatography (HPLC) in combination with mass spectrometric (MS) detection are commonly used [1,8,13,14]. However, labile cannabinoid acids cannot be analyzed separately from the cannabinoids in GC routines: the high temperatures in the injector and column cause the decarboxylation of the cannabinoid acids to their respective neutral cannabinoids [15,16,17]. Therefore, many GC methods are limited to the determination of a total THC content (=∆^9^-THC + THCA). Nevertheless, an incomplete conversion of THCA into ∆^9^-THC may cause an underdetermination of the total THC content [15]. Alternatively, derivatization mainly in the form of trimethylsilylation using *N*,*O*-bis(trimethylsilyl)trifluoroacetamide (BSTFA) and chlorotrimethylsilane (TCMS) allows a GC-differentiated determination between neutral cannabinoids and cannabinoid acids [13]. In contrast, HPLC operated at ambient temperature is suitable for the separate determination of cannabinoid acids and neutral cannabinoids (e.g., ∆^9^-THC and THCA) [13]. Limits of detection (LOD) as low as 0.03 ng/mL and limits of quantitation (LOQ) as low as 0.1 ng/mL can be obtained with HPLC-DAD-MS/MS [14]. For both methods (HPLC and GC), an appropriate sample preparation is necessary to reduce interference with the sample matrix during analysis. The sample preparation steps described in the literature are usually very similar for GC and HPLC: liquid–liquid extractions or solid phase extraction (SPE) using polar (ethyl acetate, methanol, ethanol, chloroform) or non-polar organic solvents (hexane) as well as solvent mixtures are commonly used [1]. However, sample preparation steps are usually very time-consuming. The high sensitivity with a need for considerable dilution and the narrow linear range of GC and HPLC often requires multiple replicates, because there is a large variability in the expected concentrations, resulting in considerable additional work for analysis [1,12].

A versatile method increasingly used in food chemistry for the determination of absolute contents of individual analytes in complex matrices is nuclear magnetic resonance spectroscopy (NMR) [18]. In addition to shorter measurement times compared to conventional chromatographic analytical methods, direct analysis by NMR can avoid time-consuming chemical and physical processing. This avoids possible changes in sample composition and analyte losses during sample preparation [19,20]. Furthermore, NMR methods allow simultaneous determination of multiple analytes. Despite these potential advantages, NMR methods have only been rarely employed in the determination of cannabinoids: beside the authentication of *Cannabis sativa* L. hemp varieties [21,22], ^1^H NMR has currently been described so far by only a few working groups to quantify cannabinoids in extracts of the hemp plant and hemp flowers [20,23]. The less sensitive ^13^C NMR was also used by Marchetti et al. to quantify CBD, CBDA, CBG and cannabigerolic acid (CBGA) in hemp extracts [19]. To the best of our knowledge, however, we are not aware of any work of either ^1^H NMR or ^13^C NMR methods that aim for direct quantitative determination of multiple cannabinoids in the complex matrix of CBD oils without further purification or isolating steps.

In this work, we developed a quantitative ^1^H NMR method for screening of cannabinoids such as ∆^9^-tetrahydrocannabinol (∆^9^-THC), cannabidiol (CBD), and other cannabinoids in CBD oils in order to determine toxicologically relevant levels of the analytes (especially ∆^9^-THC). We first selected a solvent suitable for CBD oils and assigned the signals of the cannabinoid pure compounds to respective protons of the analyte molecules. A NMR experiment including automated data evaluation routine was developed based on spiking experiments to identify selective cannabinoid signals in the oil matrix. The proposed method was validated and compared to the HPLC-MS/MS data of 46 commercial CBD oils.

## 2. Materials and Methods

### 2.1. Reagents and CBD Oil Samples

The deuterated solvents used for NMR were as follows: methanol-d_4_ (CD_3_OD, purity 99.8%) was purchased from Merck (Darmstadt, Germany), dimethyl sulfoxid-d_6_ (DMSO-d_6_, purity 99.9%) was purchased from Eurisotop (Saint-Aubin, France), while deuterated chloroform-d_1_ (CDCl_3_, purity ≥ 99.8%) and internal reference standard tetramethylsilane (TMS) for NMR measurement were obtained from Roth (Karlsruhe, Germany).

Crystalline cannabidiol (CBD, purity 99%) was purchased from CBDSHOP24.de (Harrislee, Germany) and crystalline cannabigerol (CBG, purity 98.3%) was purchased from cbd-brothers.de (Chemnitz, Germany). Crystalline cannabidiolic acid in the form of a dicyclohexylammonium-salt (DCHA-CBDA, purity ≥ 99%) and cannabinol (CBN, purity 99.6%) were purchased from THC Pharm (Frankfurt, Germany). Crystalline (-)-∆^9^-tetrahydrocannabinolic acid A (THCA) was obtained from Sigma Aldrich (Darmstadt, Germany). The cannabinoids (-)-∆^9^-tetrahydrocannabinol (∆^9^-THC), (-)-∆^8^-tetrahydrocannabinol (∆^8^-THC), (-)-∆^9^-tetrahydrocannabivarin (THCV) were commercially available exclusively dissolved in methanol (mass concentration 1 mg/mL; 1 mL ampule size) purchased from LGC Dr Ehrenstorfer (Teddington, UK). All commercially available reagents were used without prior purification.

The external standard substances ethylbenzene (EB) and tetrachloronitrobenzene (TCNB) were purchased from Sigma Aldrich (Darmstadt, Germany).

Commercial CBD oil samples measured in this study all originate from routine analysis of both planned and suspected samples. The samples were examined at the CVUA (Chemisches und Veterinäruntersuchungsamt) Karlsruhe in the period from March 2019 to July 2020. The declared CBD contents range from 2.75 to 30 wt.-%. Of the 46 samples tested, 39 samples were dissolved in hemp seed oil and seven samples were dissolved in medium-chain triglyceride (MCT) oil.

### 2.2. NMR Methodology

All ^1^H NMR measurements were performed using a Bruker Ascend 400 spectrometer (400 MHz) with a PA BBI 400S1 H-BB-D-05 probe head or a Bruker UltraShield 400 MHz with a 5mm PASE 1H/D-13C Z-GRD probe head, both equipped with an automatic sample changer (SampleXpress H15000-01) (all equipment from Bruker Biospin, Rheinstetten, Germany). All spectra were recorded at 300.0 K after 5 min for thermal equilibration. For sample preparation, approximately 100 ± 2 mg of the sample was dissolved in 0.6 mL solvent which contained 0.05% TMS. The resulting solution was placed for NMR measurement in a 5 mm NMR sample tube (Deutero, Kastellaun, Germany).

#### 2.2.1. ^1^H NMR Experiment and Spectral Processing for the Solvent Testing and Signal Assignments

In order to identify an appropriate solvent for the cannabinoid quantification experiment, NMR spectra of pure cannabidiol and commercial CBD oils based on hemp seed oil and MCT oil were recorded. Prior to the NMR experiments, the samples were dissolved in either pure solvent (CD_3_OD, CDCl_3_) or solvent mixtures (CDCl_3_/CD_3_OD (3:2, 2:1, *v*:*v*) and CDCl_3_/DMSO-d_6_ (5:1, *v*:*v*)). These solvents and solvent mixtures were previously proposed in the literature for NMR measurement of hemp extracts or edible oils [9,19,20,22,23,24,25]. To assign the cannabinoid signals in the ^1^H NMR spectrum of CBD oils, reference spectra of different cannabinoids were recorded in CDCl_3_ and their compound-specific coupling constants, multiplicities, and chemical shift were recorded and verified with literature data. The reference substances were selected based on the main naturally occurring cannabinoids of hemp and the recommendation of the EU Commission (EU) 2016/2115 on the monitoring of ∆^9^-THC, its precursors and other cannabinoids in foods [26]. Therefore, CBD, ∆^9^-THC, ∆^8^-THC, CBN, CBG and THCV were measured, as well as the cannabinoid acids THCA and CBDA. For signal assignment of the individual cannabinoids, one-dimensional ^1^H NMR spectra were recorded for various cannabinoids and spike solutions with and without oil matrices were performed (Table 1). In case of ∆^9^-THC, ∆^8^-THC and THCV, the nondeuterated methanol of the commercial products were removed prior to the NMR experiment: the solutions were evaporated to dryness for approximately 30–90 min at 45 °C via nitrogen evaporation depending on the quantities and redissolved with deuterated NMR solvent.

#### 2.2.2. ^1^H NMR Experiment and Spectral Processing for Quantification

The NMR spectra were recorded using an adapted composite experiment from Bruker consisting of two one-dimensional ^1^H NMR experiments (Table 2) including a pulse sequence that suppresses selected intense lipid signals in the spectrum. The lipid signals are suppressed in the frequency ranges of 347–357 Hz (0.87–0.89 ppm), 504–524 Hz (1.26–1.31 ppm), 640–650 Hz (1.60–1.63 ppm), 803–823 Hz (2.01–2.06 ppm), 919–929 Hz (2.30–2.32 ppm), 1102–1112 Hz (2.75–2.78 ppm), and 2116–2156 Hz (5.29–5.39 ppm). The multiple suppression was implemented to increase the signal intensity of minor components, e.g., the cannabinoids. Detailed descriptions of multiple suppression for oil matrix are given by Ruiz-Aracama et al. [27] and Longobardi et al. [28]. An automatic estimation of the 90° pulse width (P1) for each sample was implemented, because the effectiveness of the 90° pulse varies from sample to sample depending on their physicochemical properties [29].

In order to achieve correct signal intensities and to avoid systematic measurement errors in the NMR method, the acquisition parameters were carefully optimized during method development. The receiver gain (RG) was set to 16 after optimization via multiple suppression. The temperature effects were investigated with a CBD and ∆^9^-THC spiked hemp seed oil in CDCl_3_ between 280 K and 320 K. In addition, complete relaxation of the analyte signals and thus coherent signal intensity are necessary to avoid underestimation of the analyte concentrations. To verify the commonly chosen relaxation delay of five times the longitudinal relaxation time (T1), T1 was determined for the selected cannabinoid signals by means of an inversion recovery experiment.

The raw spectra were processed using the software TopSpin 3.2 (Bruker Biospin, Rheinstetten, Germany). The time domain was set to 131,072 data points with a spectral width of 20.5617 ppm. The size of real spectrum (SI) was extended to 262,144 by zero filling. For further spectra processing, the free induction decay (FID) was multiplied with an exponential window function to achieve a line broadening of 0.30 Hz and the spectra were automatically phase- and baseline-corrected.

#### 2.2.3. Automated Quantification Routine Using PULCON Principle

The quantification was performed using PULCON (PULse length based CONcentration determination) which is based on the use of an external standard [29,30]. The technique directly relates the measured signal intensity per proton of the analyte to the measured signal intensity per proton of a calibration standard of known concentration for determination of the absolute concentration [30].

The external standard (also called quantification reference, QR) was composed of tetrachloronitrobenzene (TCNB, 4662 mg/L) and ethylbenzene (EB, 3506 mg/L) dissolved in CDCl_3_. Note that for correct signal intensities, the device-specific response (ERETIC factor, f_ERETIC_) relies on the full relaxation with T1 as the minimum spectra acquisition time [30]. Due to the long T1 of TCNB and EB in CDCl_3_ (10.9 s and 8.8 s, respectively), empirical spectroscopic correction factors (1.30 for TCNB; 1.22 for EB) were added to the ERETIC factor-defining equation to correct the signal intensity (Equation (S1)). This enables a reduction of the measurement time to 14 s by a factor of 4 compared to complete relaxation. Due to the suppression ranges for the fatty acids, the signals at 1.24 and 2.65 ppm cannot be used for the calculation of the ERETIC factor. The factor is therefore calculated exclusively from the two signals of the respective aromatic protons. The defined signal ranges of TCNB (7.64–7.83 ppm, 1 proton, singlet) and EB (7.0162–7.2341 ppm, 3 protons, multiplet) were integrated and the average ERETIC factor was used for further calculations.

Using this modified ERETIC factor, the PULCON relation (Equation (S2)) yields the mass concentration of the analyte in (mg/L sample solution) based on NMR signal areas. Note that all experiments were recorded with the same NMR spectrometer as this relationship is only valid if both the QR and sample solutions were recorded with the same acquisition parameters. To report the cannabinoid content in (mg/kg sample), the results obtained with PULCON equation were converted to (mg/kg sample) using a simplified conversion (see Equation (S3)): (1) volume contractions that may occur during mixing of sample solution are neglected, (2) the sample weight is set to 100 mg sample for all calculations, neglecting the actual uncertainties, (3) an experimental determined average CBD oil density of 0.89 g/mL is used, neglecting any influence of the actual density of the individual carrier oil.

At the end of each measurement series, a control sample (sample with cannabinoids contents previously determined) was also measured. The results of the sample should not deviate by more than two times the standard deviation from the mean value of the precision measurements. The concentration quantification (signal integration and calculation) of the processed spectra were performed using MatLab (version 2015b, The MathWorks, Natick, MA, USA). In addition, all obtained spectra and fit graphs were visually checked to control the correctness of the MatLab routine. Cannabinoid contents whose value are below the determined LOD (see Section 3.6.2) were automated assigned to zero in the processing routine. In order to avoid or diminish the influence of the sample matrix on individual signals, the signal areas of the signals CBD at 4.63 ppm (called CBD 3 in the further work), ∆^9^-THC (δ = 6.15 ppm) and ∆^8^-THC (δ = 6.12 ppm) were determined using a line fitting algorithm.

### 2.3. Validation Studies

The developed NMR measurement program (see Table 2) was validated using the general characteristics for method performance according to the German standard DIN 17025 [31]: linearity, detection limit (LOD), quantification limit (LOQ), precision and recovery. The validation was performed using a spiking series where the cannabinoids were added to CBD oil (matrix: hemp seed oil with 10–15 wt.-% CBD) or pure hemp seed oil. The spike concentration ranges were 100–1300 mg/L for the cannabinoids CBD, ∆^9^-THC, ∆^8^-THC and CBN and additionally for CBD 8–35 g/L. For the preparation of the spiked matrix samples, 100 mg of the sample matrix was weighed into a 4 mL glass vial and freshly prepared cannabinoid standard solutions in CDCl_3_ were added in specified concentrations ranges and filled up with CDCl_3_ + TMS to a total amount of 600 µL of solvent. A detailed overview of the approaches used for method validation are presented in the Appendix A. The process parameters to be investigated were derived from the calibration series described. The LOD and LOQ were determined based on the calibration line method according to the German standard DIN 32645 [32]. For the calculation, a significance level of 0.05 was assumed for the LOD and a significance level of 0.025 was assumed for the LOQ. Recoveries were also calculated from the spiking series for each cannabinoid at different concentrations. The contents were calculated after subtracting the blank value using the PULCON method. The coefficient of variation (CV) was used as criterion for evaluating the precision of the proposed qNMR method. The precision was assessed using a selected control sample (commercial CBD oil with a declared content of 30% CBD). The precision was evaluated under intraday repeatability conditions, where the identical sample material was weighed and processed five times within a single day, and interday repeatability conditions, where the sample was weighed and processed five times on another day followed by a series of measurements on six further days. Measurement accuracy was determined on a batch of the control sample which was weighed once and measured five times. For measurement series with a normal distribution of the variances (Shapiro-Wilk test, statistical certainty 95%), the coefficients of variation (CV) were determined using OriginPro 2020 software (OriginLab Corp., Northampton, MA, USA).

The stability of the sample solution also plays an important role in the planning of the experiments and in the possible use of the method in routine analysis. Especially in the case of large measurement series, where time delays in sample measurement may occur, it must be ensured that the analytes in the sample do not change or degrade. To check the stability and to suggest a tolerable measurement period, the control sample already used for validation was measured repeatedly within a period of 60 h. The control sample was stored between the measurements in the autosampler at room temperature of approximately 21 °C. To avoid solvent losses, the sample was additionally sealed with parafilm—as with all sample tubes.

### 2.4. Reference LC-MS/MS Method

To verify the NMR screening method, the obtained ^1^H NMR results of the CBD oil samples were compared with the LC-MS/MS results of the routine analysis performed as described by Lachenmeier et al. [12]. In this method, the samples were dissolved in isooctane. Subsequently, the solution was purified by liquid–liquid extraction in methanol. In this process, the cannabinoids present in the sample are transferred to the methanol phase. The isooctane phase is removed and discarded after cooling and centrifugation. The methanolic extract obtained is then analyzed by LC-MS/MS [12]. The isocratic liquid chromatographic separation is performed on a C18 separation column (Raptor, ARC 18, 2.7 μm, 150 × 2.1 mm) with a mobile phase of 25% formic acid (0.1%) and 75% formic acid (0.1% in acetonitrile or methanol) and a flow rate of 0.3 mL/min and a column temperature of 35 °C. Mass detection was performed on a triple quadrupole mass spectrometer and the analyte content is calculated based on deuterated internal standards, such as ∆^9^-THC-d_3_ and CBD-d_3_ [12].

## 3. Results and Discussion

### 3.1. Influence of NMR Solvents

Considerably different solubility and chemical stability properties of cannabinoids were found within the investigated pure solvents and mixtures.

CD_3_OD was not suitable for the NMR measurement of CBD oils because the sample matrix consisting of an edible oil did not dissolve. The observed phase separation can be attributed to the low solubility of the lipids and their hydrophobic character. CBD crystals, in contrast, dissolved very well in CD_3_OD and no changes in their NMR spectrum were found in repeating NMR measurements after 3 weeks. Compared to CDCl_3_, CBD spectra in CD_3_OD showed also better resolution of the signals between 0–3 ppm and the aromatic protons at about 6.2 ppm were found to be much narrower than in CDCl_3_ (see Appendix A). No OH groups were observed with the solvent CD_3_OD. This is in line with the fast proton-deuterium exchange of polar solvents [25,30] and constitutes an advantage for qNMR: These signals are often very broad in NMR spectra and in addition to the pH, the chemical shift also depends strongly on temperature [29] impeding reliable quantification.

CBD oil dissolved very well in CDCl_3_. No changes were observed in the NMR spectra of oil containing samples (one CBD oil and one hemp seed oil spiked with CBD) within the first 24 h, but small changes were observed within 5 days. However, the spectrum of pure CBD in CDCl_3_ already showed significant changes after 12 h. Since no purification protocol was applied, the difference of pure CBD and CBD oils potentially derives from impurities in the CDCl_3_. Chloroform is known to form HCl, Cl_2_ and phosgene (COCl_2_) when exposed to air and light [33]. The higher stability of analytes in an oil matrix, compared to pure chloroform, is potentially linked to the unsaturated fatty acids and tocopherols which are natural antioxidants, i.e., molecules scavenging free radicals [34].

CBD oil samples dissolved completely in a solvent mixture of CDCl_3_ and CD_3_OD at both investigated mixing ratios (3:2 and 2:1, *v*:*v*). The polar CD_3_OD ensures that the polar compounds present in the sample pass easily into the solution, e.g., cannabinoid acids and polyphenols. No significant differences were found by dilution and in NMR spectra between the two different mixing ratios. However, the pure CBD and CBG in CDCl_3_/CD_3_OD (2:1, *v*:*v*) already showed a significant reduction of the signal areas after 12 h. In the spectra of the pure cannabinoids CBD and CBG as well as in the CBD oil samples, the relative signal area ratio of the signal of the two aromatic protons (H-3′+ 5′) were 0.05 with a CDCl_3_/CD_3_OD mixture. This is 40 times less than the expected nominal value of 2.0. In contrast, the nominal proton numbers were matched using pure CD_3_OD or pure CDCl_3_. For example, the relative proton ratio of CBD was 1.9563 (pure CDCl_3_) and 1.9933 (pure CD_3_OD), respectively. Incomplete relaxation can be ruled out as a cause of the decreased signal intensity. The signal intensity is stable above a relaxation time of 30 s. However, the proton ratio is almost invariant between 0.0317 and 0.0359 over the entire measurement range of 4–60 s.

The solvent mixture of CDCl_3_ and DMSO-d_6_ (5:1, *v*:*v*) proved not to be suitable for the NMR spectroscopy of CBD oils. The CBD signals between 3–7 ppm were completely masked by the sample matrix, making it impossible to evaluate CBD signals.

Among the investigated solvents and mixtures thereof, pure CDCl_3_ showed excellent solvent properties for a qNMR method because the aromatic protons can be properly resolved in the NMR spectra. Beyond the spectra quality, CDCl_3_ is efficient for routine analysis due to several reasons: (1) the CDCl_3_-diluted samples maintain a stability for days, (2) the compound is commercially available, and no mixture needs to be prepared and (3) it is relatively inexpensive compared to other deuterated solvents. Consequently, CDCl_3_ was chosen as solvent and all results described hereafter exclusively refer to this solvent.

### 3.2. Signal Assignments of Cannabinoids

An overview of ^1^H NMR spectra of pure cannabinoids in CDCl_3_ is shown in Figure 2 and detailed signal assignments of the cannabinoid signals are given in the Appendix A. The signal labels in the following refer to the numbering of the protons according to Figure 1.

The ^1^H NMR spectra showed nearly identical coupling and chemical shifts for the cannabinoids as described by Choi et al. [9]. After normalizing the signal intensities of the easily recognizable triplet at about 0.89 ppm (CH_3_ group at the C_5_ or C_3_ alkyl side chain) to three protons, appropriate numbers of protons as well as correct proton ratios were obtained for the cannabinoid signals.

The NMR spectra of the individual cannabinoids, in general, showed only minor differences compared to each other (see Figure 2), which reflects the structural similarities of the cannabinoids. Especially in the high field of the spectrum (approximately 0–3 ppm) where strongly shielded protons are mainly detected (e.g., alkyl protons), the ^1^H NMR spectra of the cannabinoids showed strongly overlapped signals. In this region, the proton signals of the C_5_ or C_3_ alkyl side chain of the cannabinoids emerge. For example, at a chemical shift of about 0.89 ppm, the signal of the CH_3_ group (H-5″) of the respective C_5_ side chain is followed by the CH_2_ group signals (H-4″ and H-3″) at about 1.30 ppm and the signal of the CH_2_ group at position H-2″ at about 1.56 ppm. In addition, an overlap with water residues from the solvent or substance is often observed in this region. Water residues exhibit a singlet at 1.56 ppm in chloroform [35].

In contrast, the chemical shifts of the signals between 3–7 ppm differ significantly between the individual cannabinoids (see Figure 2). In this region, the olefinic and aromatic proton signals or less strongly shielded signals are generally to be found. Because of these different chemical shifts in the individual cannabinoids, this range between 3–7 ppm is better suited for distinguishing the cannabinoid spectra, but the differences in chemical shift are still very small, which impairs cannabinoid differentiation.

The compounds ∆^9^- and ∆^8^-THC are representative examples to illustrate the difficulties in the spectroscopic differentiation of cannabinoids: structurally, ∆^9^-THC and ∆^8^-THC differ in the position of their double bond (cf. Figure 1). Thus, the expected differences in the NMR spectra are also minimal, e.g., the corresponding signals at position H-10 (about 1.09 ppm), H-9 (about 1.40 ppm) and H-7 (about 1.69 ppm) differ in their chemical shift only by about 0.1 ppm (cf. Appendix A). Additionally, it is mainly proton signals which differ in their chemical shift, corresponding to location directly or adjacent to the double bond in the molecule, e.g., positions H-1, H-2, H-4. But the position H-3′ on the aromatic ring shows differences with δ = 6.14 ppm for ∆^9^-THC and δ = 6.10 ppm ∆^8^-THC. The cannabinoids ∆^9^-THC and THCV differ only minimally and their differences are even smaller compared to ∆^9^-THC and ∆^8^-THC: the CH_2_ groups (H-4″ + H-3″) at approximately 1.32 ppm and minor shifts in the signals corresponding to the H-5″ and H-2″ positions are completely missing in case of ∆^9^-THC and THCV.

Another unique feature of the cannabinoid CBD is that the proton signal of the aromatic protons (see Figure 2, CBD, H-3′ + 5′ at about 6.2 ppm) clearly broadens in the ^1^H NMR spectrum in contrast to the analogous signals of the other cannabinoids. In addition, the chemical shifts of these signals were strongly temperature-dependent (see Figure 3). This observation has already been described by several research groups [9,20,36]. The effect of this broadening is attributed in the literature to restricted rotation of the single bond between the phenyl carbon C-1′ and the C-1 carbon of the terpene moiety [9,36]. The use of a protic solvent such as deuterated methanol can prevent the broadening [9].

### 3.3. Signal Identification for Quantification

Quantitative NMR methods require baseline separated signals in the best case. The example of the spiked hemp seed oil (Figure 4) demonstrates that the ranges of approximately 0.8–3.0 ppm, 4.1–4.3 ppm and 5.2–5.4 ppm are strongly influenced by the dominant triglyceride signals of the edible oil. This again confirms that the spectral range from 0–3 ppm is not suitable for defining selective cannabinoid signals because this spectral region features intense lipid signals, overlapping the cannabinoid signals. Moreover, the spectrum also shows an offset and the signal patterns of the cannabinoids are very similar in this range.

In contrast, the region in the spectrum above 4 ppm shows significantly less overlap of the cannabinoid signals with components of the sample matrix. The signals identified in this region are marked in Figure 4. CBD features a broad signal at about 6.2 ppm, a singlet at 5.56 ppm and three multiplets (4.52 ppm, 4.63 ppm, 3.88 ppm). The three multiplets show almost baseline separation and, therefore, they were used for the quantification of CBD in this work.

The ∆^9^-THC is identified in the spectrum by a doublet at 6.15 ppm and a quintet at 6.33 ppm. All other signals are influenced or overlapped by the sample matrix. However, the cannabinoid THCV shows identical signals as ∆^9^-THC. Therefore, an integration of the signals results in the sum of ∆^9^-THC and THCV. Fiber hemp usually contains less than 0.6 mg THCV and about 200 mg ∆^9^-THC/g dry weight in its flowers [37]. We note that THCV also plays a negligible role from a consumer protection point of view as THCV exhibits psychoactive effects with a potency four to five times lower than ∆^9^-THC [38]. The expected THCV content in the hemp plant is more than one hundred times lower than the ∆^9^-THC which presumably places THCV well below the LOD and, thus, THCV is neglected in our qNMR method. The subsequent quantification of ∆^9^-THC was performed using the doublet clearly visible at a chemical shift of 6.15 ppm. For the quantification of ∆^8^-THC, the doublet at 6.12 ppm was selected.

Several signals could be directly assigned to CBN in the spectrum. CBN shows a total of four doublets at 6.31 ppm, 6.41 ppm, 7.05 ppm and 7.13 ppm and one singlet at 8.21 ppm. Due to the high density of cannabinoid signals between 6 and 7 ppm and the associated difficulty in assignment, only the signals 7.05 ppm, 7.13 ppm and 8.21 ppm were used for quantification purposes.

Many cannabinoids created signals between 6.10–6.28 ppm on a broad hump (see Figure 4). The broad hump is assigned to the aromatic proton signal of CBD. This applies to ∆^9^-THC and ∆^8^-THC, but also the signals of the cannabinoids CBG (δ = 6.24 ppm), CBDA (δ = 6.23 ppm) and THCA (δ = 6.228 ppm) are located in small distances on the top of the broad CBD hump. Measurements showed that the chemical shifts of the cannabinoids slightly shift as function of the CBD content. As shifting signal positions require more sophisticated evaluation routines, these signals were also excluded for quantitative determination.

We further note that the signals of OH groups are not suitable for quantitative determination with direct signal integration strategy applied in this work. These signals are particularly shifted into the low field of the spectrum (e.g., THCA: 12.31 ppm, CBDA: 11.98 ppm) due to their low shielding. The sample matrix has little influence on the spectrum in this range. However, the qualitative observation of various CBD oil samples showed that the chemical shifts of the OH signals largely varies, impeding a reliable automated signal picking and integration. The observed strong chemical shifts are in line with the temperature- and pH-dependence of OH protons [30]. Moreover, OH groups are generally considered to be problematic for quantification because of the possible proton–deuterium exchange [30].

### 3.4. Optimization of NMR Measurement Protocol

#### 3.4.1. Receiver Gain and Signal-to-Noise Ratio

An optimal receiver gain (RG) is essential in NMR spectroscopy: on the one hand, a high RG leads to an overflow with baseline distortion; on the other hand, a low RG causes weak signal intensities and thus inferior signal-to-noise (S/N) ratio up to complete loss of the signals [29]. Especially, the signals of minor components in the sample, i.e., the cannabinoid signals in the edible oil in case of CBD oil, are strongly influenced by the matrix. This is because the triglycerides significantly shape the NMR spectra with their typical signal pattern (see Section 3.3). The multiple suppression of the matrix signals improved the RG from 5.6 (without suppression) to 16 (with suppression). This is in the range of desired RG (above 16–20, depending on purpose) for a highly stable S/N ratio; the S/N ratio behaves like a saturation curve as function of RG [29]. We again note that a low-resolution ^1^H NMR spectrum is recorded before the actual NOESY suppression program to sample the suppression regions for the main experiment; otherwise, not only is a single range suppressed by irradiating a suppression frequency, but all signal regions of the lipid signals are directly suppressed. The effects of the multiple suppression on the intensity of the spectra are visualized in the Appendix A.

#### 3.4.2. Temperature

Increasing the temperature from 280 to 320 K leads to a sharpening of the two aromatic protons and the close OH group (about 5.95 ppm) (see Figure 5). The temperature has a particularly strong effect on the overlapping of the two aromatic protons of the CBD molecule with the signals of other cannabinoids ∆^8^-/∆^9^-THC, affecting the signal resolution. Compared to 300 K (standard operation temperature), reducing the temperature to 280 K causes complete overlapping of the CBD with the ∆^9^-THC signal. In contrast, a measuring temperature of 320 K improved the resolution of the cannabinoid signals. However, measurements at 320 K are not preferable due to the high volatility of the deuterochloroform and its low boiling point (335 K [30]). Moreover, other temperatures than 300 K required additional measurement time for thermal equilibration which is not favored for a rapid screening method. Therefore, 300 K was identified as the optimum measurement temperature.

### 3.5. Quantification of Cannabinoids

Quantification using the PULCON method is based on the signal areas in the NMR spectrum. Consequently, numeric integration of the signals is crucial for correct cannabinoid contents. For rapid screening, the signals were integrated in the MatLab routine using curve fitting algorithm and fixed integration ranges, as shown in Appendix A.

For the integration of the signals of ∆^9^- and ∆^8^-THC (6.15 ppm and 6.12 ppm) and CBD (4.53 ppm), their overlap with the sample matrix and other cannabinoid signals must be considered. By baseline correction of the individual signal, overlaps with other signals can be reduced (see Figure 6).

However, we observed that approach of fixed integration limits for the individual signals are limited with respect to the dynamics of the broad variable CBD signal and the shifting of the ∆^9^-THC and ∆^8^-THC signals. Therefore, no general integration limits could be established for these two cannabinoids. What is more, the chemical shift of ∆^9^-THC (6.15 ppm) and ∆^8^-THC (6.12 ppm) differ only slightly causing an unreliable signal assignment: manual checking of the automated fits revealed that fit algorithm interchanged ∆^9^-THC/∆^8^-THC signals in some cases.

We found that nearly all of the samples where swapping occurred were MCT oil-based CBD formulations. Consequently, other integration areas were implemented for MCT oil-based CBD oil in the MatLab script to prevent the swapping.

### 3.6. Method Validation

#### 3.6.1. Linearity

The regressions between the absolute analyte signal integrals and target concentrations showed a linear relationship (correlation coefficient of >0.997) of the spiking series of the respective cannabinoid in hemp seed oil or CBD oil 10–15 wt.-%. The procedural variance coefficients were all below 5%. The values scatter within an acceptable range around the respective linear fit and the influence due to random errors is low. Therefore, a linear correlation between proton signal and concentration could be confirmed for all selected cannabinoid signals in the concentration range of about 100–1300 mg/L, and additionally for CBD from 8–37 g/L. The calibration data for the investigated cannabinoids are documented in the Appendix A.

#### 3.6.2. Analytical Limits

In order to obtain quantitative results as precisely as possible with NMR, the S/N ratio of the signal used for quantification should be at least 250 [28]. An estimation of the S/N ratio based on the spectra obtained from the spiking series was not possible because the sample matrix leads in part to a strong offset. However, hemp seed oil, which is mainly the matrix of CBD oils, cannot be used as a blank sample because it contains naturally small amounts of cannabinoids [1,39]. Therefore, the LOD and LOQ were calculated based on the calibration line method according to the German standard DIN 32645 (see Section 2.3). The calculated limits are listed in Table 3.

In a previous study by Lachenmeier et al., HPLC measurements showed that, first, almost all CBD oils tested exceeded toxicity thresholds of European or German guidelines in relation to the manufacturer’s recommended daily dose and, second, THC levels in CBD oils varied widely [12]. With respect to our qNMR method, 45% of the samples of this previous study would have exceed the LOD (608 mg/kg) and 15% the LOQ (1858 mg/kg). This shows that the daily THC dose is highly dependent on the manufacturer’s information about the maximum daily intake of the products. Thus, even CBD oils with small amounts of THC can cause THC exceedance if the daily dose is high enough or exceeded by the consumer. Such CBD oil cannot currently be qualitatively recorded by NMR. Nevertheless, control of products with extreme levels is possible, e.g., to avoid acute toxicity from THC contents exceeding the lowest observed adverse effect level (LOAEL) of 2.5 mg/day [11] or psychotropic levels (i.e., 5–15 mg/dose) [12].

#### 3.6.3. Recovery

The recoveries for CBD and CBN were 114–120% and 117–135%, respectively, for the investigated signals in the entire concentration range. The recovery thus deviated almost equally upwards over the entire concentration range which indicates a systematic deviation. However, recovery tests of a CBD calibration series, which were measured with the same NMR measurement program without matrix components, also showed too high recoveries in the same order of magnitude (results see Appendix A). This suggests that factors other than the matrix could be responsible for the high recoveries and the systematic deviation.

In contrast, the recoveries of ∆^8^-THC and ∆^9^-THC improved with increasing concentration of the calibration points. While the recoveries were 57% (∆^8^-THC) and 35% (∆^9^-THC) at a concentration of 140 mg/L (calibration point 1), recoveries of 93% (∆^8^-THC) and 84% (∆^9^-THC) were obtained for 1260 mg/L (calibration point 5). The baseline correction has a strong effect due to the CBD overlap of both signals. This effect is visible especially at low concentrations, close to the LOD.

The recoveries for CBD, CBN as well as ∆^9^- and ∆^8^-THC determined in the validation experiments were accepted for the developed qNMR method. A further check of the accuracy of the qNMR method is performed in Section 3.7 by comparing the sample results of commercial CBD oils with an independent, validated method. All recoveries obtained are summarized in Appendix A.

#### 3.6.4. Precision and Stability

The results showed that the coefficient of variation (CV) is of the order of 1% for intraday repeatability conditions and of the order of about 3% for interday repeatability conditions (see Table 4). No CVs could be determined for the CBN signals because their levels in the sample were below their LOD. The required reproducibility and precision of the method is generally based on the requirements and measurement task. While a reproducibility of less than 2% is usually required for methods in the pharmaceutical field, coefficients of variation of approximately 5–10% are quite acceptable in other fields [40]. In the literature, an inaccuracy below 2% is generally reported for qNMR methods [29]. The reproducibility of the determination corresponds to the current state of the art for the investigated signals and thus was accepted. Compared to the repeatability determined, the results obtained are lower. This is in line with our expectations, as usually sample preparation and sampling have a dominating effect on the result compared to the actual measurement. Within the measuring period of 60 h, no significant signal change or new signals were observed, which would indicate degradation reactions. Thus, the signal areas within the tested period are within the range of the determined measurement uncertainty. Due to the high volatility of the solvent, chloroform samples should be measured within 24 h.

### 3.7. Applicability for Cannabinoid Screening of Commercial CBD Oils

Comparing the results of the derived qNMR method to the standard LC-MS/MS method showed a linear correlation for ∆^9^-THC (see Figure 7). What is more, the comparison indicates that most of the samples are below the NMR(LOQ) (1858 mg/kg). Formally, the analyte ∆^9^-THC can therefore only be assessed qualitatively by qNMR, but not quantified with sufficient confidence. However, we found that the qNMR method also provides acceptable quantitative measured values below the LOQ as outlined in the following.

The automatic qNMR method only output values above the LOD of the NMR (608 mg/kg, see Figure 7). The LC-MS/MS results provided a ∆^9^-THC content below the LOD of the NMR for a total of 23 samples. Moreover, concentrations below the LOD of the NMR cannot be qualitative determined by the qNMR method. The data of the samples whose LC-MS/MS results are above the NMR(LOD) indicate a linear correlation between both methods (Figure 7). Only sample 16 (NMR: ∆^9^-THC n.n.; LC-MS/MS: 8520 mg/kg ∆^9^-THC) was significantly underdetermined but sample 25 (NMR: 3629 mg/kg ∆^9^-THC; LC-MS/MS: 748 mg/kg ∆^9^-THC; recovery = 485%) was overdetermined.

A linear regression (model type: y = a x + b) between the NMR and LC-MS/MS results was performed for all samples whose ∆^9^-THC content exceeded the LOD of the NMR method to assess the trueness. The two highly deviant samples (Figure 7) were considered as outliers and were not included in the linear regression, reducing the number of data points to 13. The slope (a = 0.91254) as well as the correlation coefficient (R = 0.92974) indicate a good agreement of both methods and thus reasonable trueness in the investigated concentration range (Appendix A). We note that the intercept (b = 38.60133) is rather high compared to the LOD (NMR). The residuals of the linear regression analysis show a uniform scattering. A *t*-test rejected systematic differences between the NMR and LC-MS/MS method with a probability of 95%. The relative deviations of the qNMR results compared to the LC-MS/MS results range from 56–128%. The deviations potentially originate from very strong adjustments that are necessary for the integration of the ∆^9^-THC signal due to the overlapping with the CBD signal and corresponding baseline shifts.

The determination of CBN in commercial CBD oils showed that the CBN contents usually contained in samples with less than 423 mg/kg (HPLC results) are mostly below the detection limit of at least 504 mg/kg determined for NMR. In addition, the variances within the three signals were very large and showed in part strong offsets, which indicate an influence by the sample matrix. Therefore, the developed method is not suitable for a qualitative detection of CBN.

As CBD reference values were only available for a few samples (as a result of difficulty to hit the calibration range in the LC-MS/MS), no correlation between the two methods could be derived with sufficient confidence. However, the results obtained for CBD by this qNMR method showed promising correlation with the LC-MS/MS results: these few results agreed with their recoveries of 82% and 105% (two samples) very well. For this purpose, the developed qNMR method could offer the possibility to derive the sample dilution necessary for the LC-MS/MS in advance to increase the analysis throughput in the future. We note that, during the evaluation of CBD, the proton signal CBD 2 (H-9 *cis*, 4.53 ppm) tended to yield a higher CBD content than the other two signals CBD 1 (H-1, 3.88 ppm) and CBD 3 (H-9 *trans*, 4.63 ppm). This was unexpected given a constant proportionality between number of protons and signal intensity, but the prerequisite for constant proportionality is that none of the signals is influenced by matrix. Therefore, only the two signals CBD 1 and CBD 3 were used for the evaluation.

## 4. Conclusions

In this study, we assigned and identified selective proton signals of the cannabinoids CBD(A), CBN, CBG, ∆^8^-THC, ∆^9^-THC(A), THCV within ^1^H NMR spectra of commercial CBD oils. Signals suitable for the quantification were found in the region with chemical shifts between 3–8 ppm. Based on the signal assignment, a direct quantitative NMR measurement method for the screening of cannabinoids in CBD oil was developed. The method proved to be sufficiently linear, precise and correct for the cannabinoids CBD, ∆^9^-THC, ∆^8^-THC and CBN during the validation performed. The PULCON method allowed the quantification of cannabinoids without the use of cannabinoid reference substances, which is a special advantage in this case because some of the pure substances fall under narcotics regulations and require special efforts for purchase and storage. Due to the implemented multiple suppression of triglyceride signals and the associated simple processing—consisting of a direct measurement of the sample in chloroform and the MatLab-based evaluation—the method could be established in a very resource-efficient way. This enables an efficient workflow when the method is potentially used in routine analysis by laboratories of official food monitoring.

The developed ^1^H NMR method was verified using a series of commercial CBD oils and proved to be efficient for the determination of ∆^9^-THC. The obtained ∆^9^-THC levels above the detection limit (LOD = 608 mg/kg) showed a linear relationship to the analogous LC-MS/MS results with recoveries between 56–120%. The LOQ of 1858 mg/kg is above the contents of most CBD oil samples (range around 40–3300 mg/kg). Thus, the method is predominantly suitable to make qualitative statements about the ∆^9^-THC content of CBD oils. Conspicuous samples can thus be identified by the qNMR method in the sense of a screening. However, it is necessary to further lower the LOD and LOQ of the current method in order to detect a larger proportion of samples to be objected. Especially in the case of very high ∆^9^-THC contents, which entail a possible exceeding of the ARfD of EFSA, the method is furthermore suitable to reliably determine the magnitude of the CBD content. In order to verify a possible quantitative correlation between qNMR and LC-MS/MS for CBD, more LC-MS/MS result are needed. For cannabidiol, the NMR method was able to show good agreement with the LC-MS/MS method with recoveries of 82% and 105%, in spite of only a few results. In contrast, the current method was not suitable for the quantitative determination of cannabinol in CBD oils, as its content is usually below the determined LOD of approximately 500 mg/kg.

In summary, the developed screening method can be a useful addition to the existing analysis procedure. By screening the sample prior to the LC-MS/MS routine, suitable calibration ranges or optimal sample dilution for the LC-MS/MS analysis can be easily determined. This avoids time-consuming repetitions of the determinations.

## Figures and Tables

**Figure 1 toxics-09-00136-f001:**
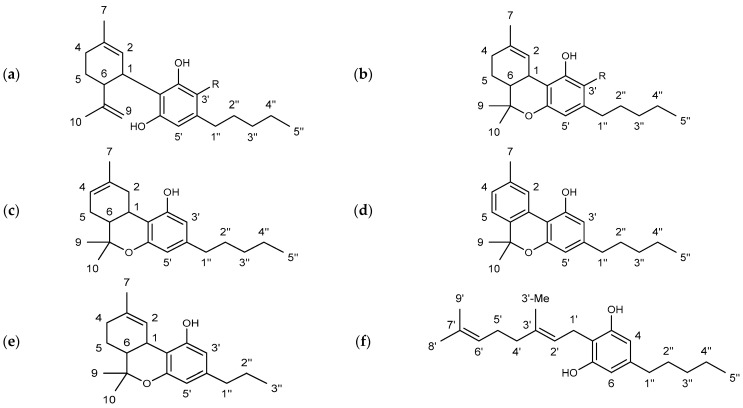
Chemical structures of the main cannabinoids studied in this work including the applied numbering system. (**a**) R = H: cannabidiol (CBD), R = COOH: cannabidiolic acid (CBDA); (**b**) R = H: ∆^9^-tetrahydrocannabinol (∆^9^-THC), R = COOH: ∆^9^-tetrahydrocannabinolic acid A (THCA); (**c**) ∆^8^-tetrahydrocannabinol (∆^8^-THC); (**d**) cannabinol (CBN); (**e**) ∆^9^-tetrahydrocannabivarin (THCV); (**f**) cannabigerol (CBG).

**Figure 2 toxics-09-00136-f002:**
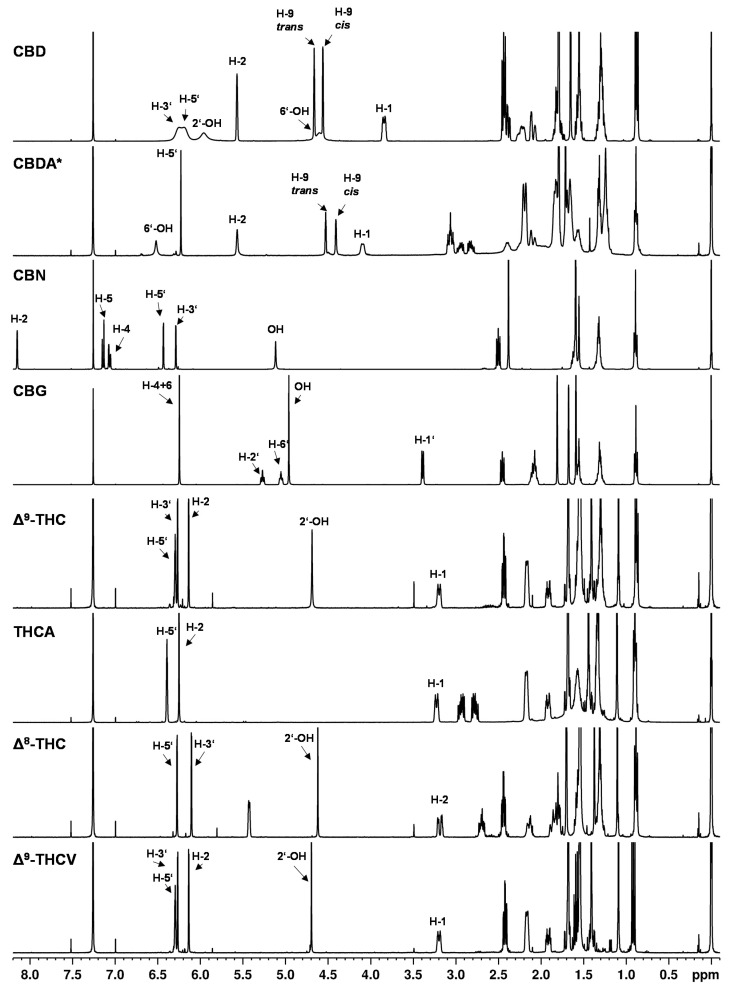
^1^H NMR spectra of cannabinoids CBD, CBDA, CBN, CBG, ∆^9^-THC, THCA, ∆^8^-THC, THCV dissolved in CDCl_3_. * Cannabinoid stabilized as *N*,*N*-dicyclohexylammonium salt.

**Figure 3 toxics-09-00136-f003:**
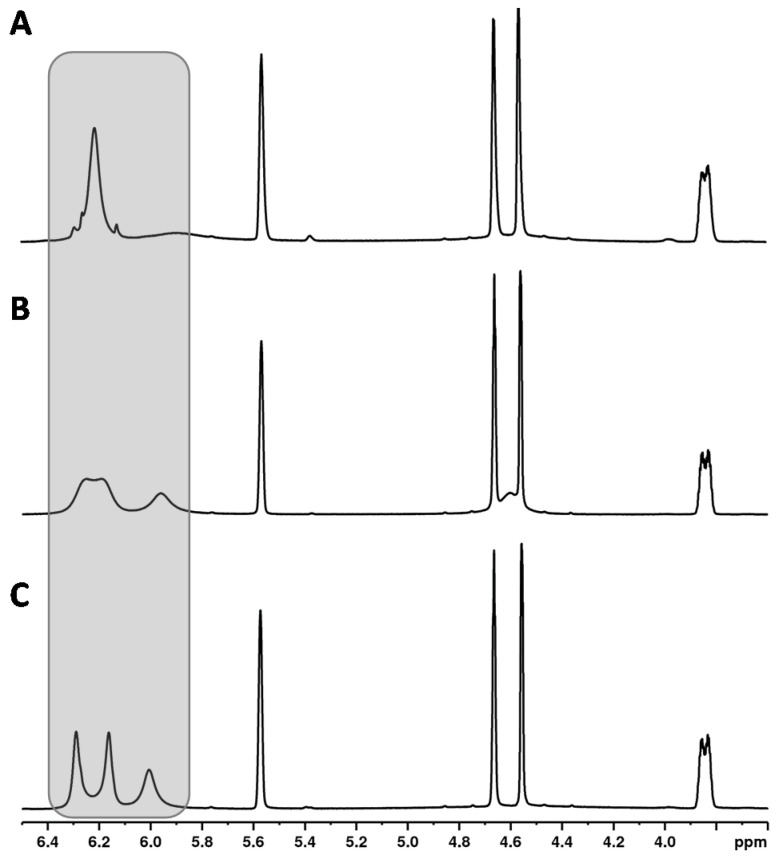
^1^H NMR spectra of CBD recorded at (**A**) 315 K, (**B**) 300 K, (**C**) 285 K.

**Figure 4 toxics-09-00136-f004:**
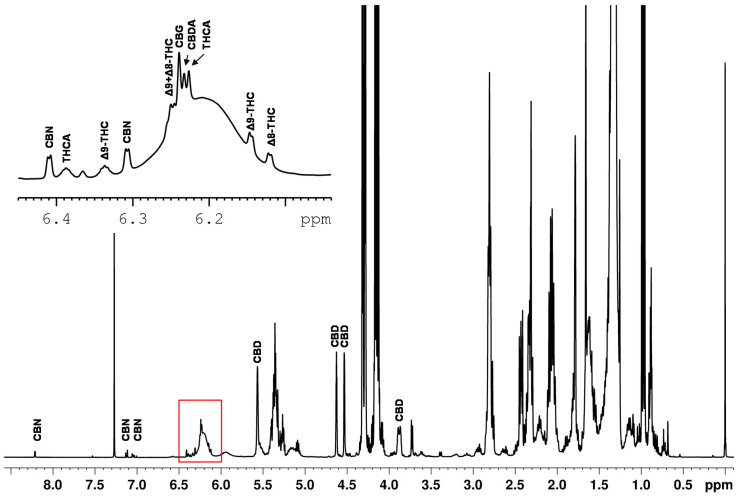
^1^H NMR spectrum of a hemp seed oil spiked with CBD, CBDA, ∆^9^-THC, ∆^8^-THC, THCA, CBG and CBN.

**Figure 5 toxics-09-00136-f005:**
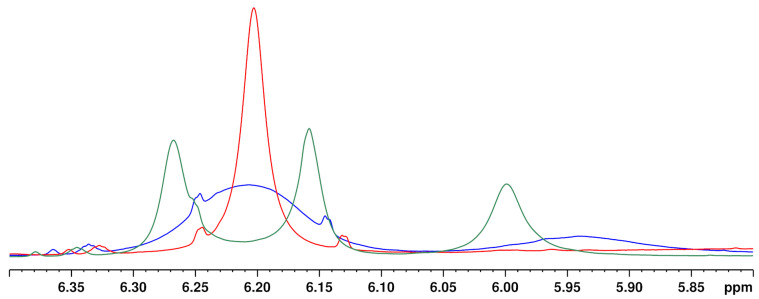
^1^H NMR spectra of a CBD + ∆^9^-THC standard in hemp seed oil in the aromatic proton region at 280 K (green), 300 K (blue) and 320 K (red). Solvent: CDCl_3_.

**Figure 6 toxics-09-00136-f006:**
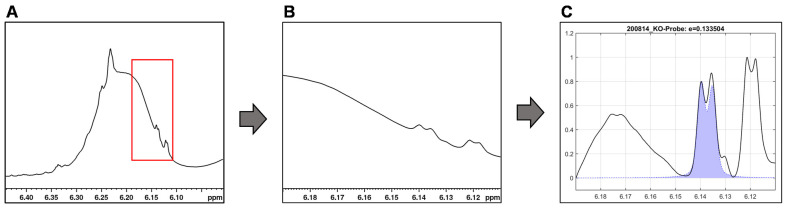
Integration by mathematical curve fitting in MatLab using the example of the ∆^9^-THC signal. (**A**) ∆^8^/∆^9^-THC overlapped by the CBD hump with region of interest highlighted in red. (**B**) Extracted region of interest. (**C**) Baseline correction and signal fit (purple filled curve) by searching a defined signal range for specific coupling and maximum for minimal matrix interference.

**Figure 7 toxics-09-00136-f007:**
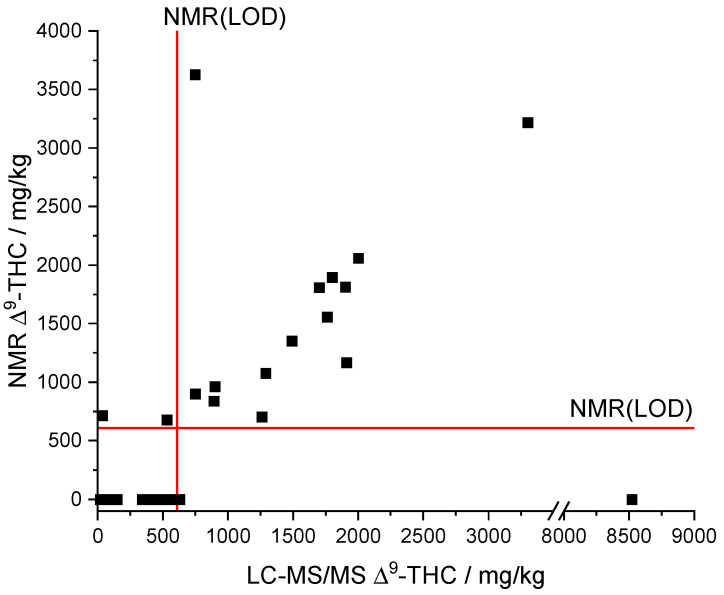
Comparison of NMR and LC-MS/MS results of the measured CBD oil samples. LOD(NMR), limit of detection of qNMR method; ∆^9^-THC, ∆^9^-tetrahydrocannabinol.

**Table 1 toxics-09-00136-t001:** Acquisition parameters of the initial NMR experiments for method development.

Parameter	^1^H NMR Experiment
Bruker pulse program name	zg
temperature [K]	300
data points	131,072
pulse [µs]	8.2
relaxation delay (D1) [s]	30
acquisition time (AQ) [s]	3.9845889
dummy scans (DS)	2
scans (NS)	8
spectral width (SW) [ppm]	20.5504
receiver gain (RG)	5.6 (oil matrix); 90.5 (without oil)

**Table 2 toxics-09-00136-t002:** Acquisition parameters of the developed multiple suppression NMR experiment.

Parameter	^1^H NMR Experiment	^1^H Multiple Suppression Experiment
Bruker pulse program name	zg	Noesygpps1d.comp2
temperature [K]	300	300
data points	65,536	131,072
pulse [μs] ^a^	about 8	about 8
relaxation delay (D1) [s]	4	6
acquisition time (AQ) [s]	3.9845889	7.9691777
dummy scans (DS)	4	4
scans (NS)	16	64
spectral width (SW) [ppm]	20.5617	20.5617
receiver gain (RG)	4	16

^a^ automatic pulse estimation.

**Table 3 toxics-09-00136-t003:** LOD and LOQ of the ^1^H NMR method for the screening of cannabinoids in CBD oils.

Analyte	Signal	δ [ppm]	LOD [mg/kg Sample]	LOQ [mg/kg Sample]
CBD	H-1	3.88	346	1092
	H-9 *cis*	4.52	134	445
	H-9 *trans*	4.63	307	979
∆^9^-THC	H-3′	6.15	608	1858
∆^8^-THC	H-3′	6.12	250	816
CBN	H-2	7.05	517	1604
	H-5	7.13	623	1897
	H-4	8.21	504	1568

**Table 4 toxics-09-00136-t004:** Coefficient of variation (CV) and measurement precision (N = 5 measurements) of different cannabinoid signals.

Signal	CBD 1	CBD 2	CBD 3	∆^8^-THC	∆^9^-THC
CV, intraday repeatability conditions (%)	1.0	1.0	0.9	1.0	1.1
CV, interday repeatability conditions (%)	2.7	2.8	3.0	2.2	3.4
CV, measurement precision (%)	0.2	0.3	1.0	0.8	0.5
CV, 60 h (%)	2.3	1.6	1.5	1.7	5.1

## Data Availability

The data presented in the method development and validation are available on request from the corresponding author. The data on official samples from the applicability study are not publicly available due to government policy.

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
