# Peer review of "A Quantitative ^1^H NMR Method for Screening Cannabinoids in CBD Oils"

_toxics, 2021, doi:10.3390/toxics9060136_

Round 1

Reviewer 1 Report

The article toxics-1241233 describes the development of the qNMR method for the determination of various cannabinoids in CBD oils. The article describes a very thoroughly complete process of the method development, although often it is at the expense of the comprehensibility of the text. Sometimes less is more. Maybe, then it can be avoided statements like that chlorine radicals are thou reactive species in chloroform solution. If you do not irradiate your NMR tubes with UV light, I have a doubt that you will find so many chlorine radicals that can cause “significant changes” in 12 h. On the other hand, if you do not perform any “purification” protocol with CDCl3, you might expect presence of DCl in your solvent.

From the description in the text it seems to be pretty clear and in line with what one can expect.

In overall, it is a solid peace of work.

Author Response

The article toxics-1241233 describes the development of the qNMR method for the determination of various cannabinoids in CBD oils. The article describes a very thoroughly complete process of the method development, although often it is at the expense of the comprehensibility of the text. Sometimes less is more. Maybe, then it can be avoided statements like that chlorine radicals are thou reactive species in chloroform solution. If you do not irradiate your NMR tubes with UV light, I have a doubt that you will find so many chlorine radicals that can cause “significant changes” in 12 h. On the other hand, if you do not perform any “purification” protocol with CDCl3, you might expect presence of DCl in your solvent.

RESPONSE: The section about solvent selection was completely revised considering the reviewer’s suggestions.

From the description in the text it seems to be pretty clear and in line with what one can expect.

In overall, it is a solid peace of work.

RESPONSE: Thank you for your assessment of our article!

Reviewer 2 Report

The manuscript presents a quantitative 1H-NMR method for screening Cannabinoids in CBD oils.

This document it reads okay. For instance, needs to be change:

- Line 2: In the title, you use the abbreviation 1H-NMR and in the rest of the document you use 1H NMR, please standardize it.

- Line 28: Please, use the same abbreviation for the proton NMR that used in the entire document. Please, use superscript for number 1 of 1H NMR.

Please give the following information or restructure it:

  • In figure 1, you identify the protons position for structures a, b and f. To follow some parts of the document the protons position for structures c, d and e are required, may be you can add it.
  • In lines 195 and 196 you give the frequency ranges where the lipid signals are suppressed. But when you identify the signals of the different structures you give the chemical shift. To compare better, may be you could give the ranges where the lipids signals are suppressed using the chemical shift.
  • You show different figures (2, 3, 4 and 5) with the 1H-NMR identifying the protons but you do not indicate in a structure that protons are. May be you can indicate it in figure 1 adding colors that you can use when identify the protons in the different spectra.
  • In section 3.1 you speak about the influence of NMR solvents used. The explanation is clear, but to follow better, may be you can add the spectra in the supplementary materials.

Author Response

The manuscript presents a quantitative 1H-NMR method for screening Cannabinoids in CBD oils.

This document it reads okay. For instance, needs to be change:

- Line 2: In the title, you use the abbreviation 1H-NMR and in the rest of the document you use 1H NMR, please standardize it.

RESPONSE: Done.

- Line 28: Please, use the same abbreviation for the proton NMR that used in the entire document. Please, use superscript for number 1 of 1H NMR.

RESPONSE: Done.

Please give the following information or restructure it:

In figure 1, you identify the protons position for structures a, b and f. To follow some parts of the document the protons position for structures c, d and e are required, may be you can add it.

RESPONSE: In Figure 1, the proton positions for the structures c, d and e were added.

In lines 195 and 196 you give the frequency ranges where the lipid signals are suppressed. But when you identify the signals of the different structures you give the chemical shift. To compare better, may be you could give the ranges where the lipids signals are suppressed using the chemical shift.

RESPONSE: Frequency ranges were expanded by the data in ppm.

You show different figures (2, 3, 4 and 5) with the 1H-NMR identifying the protons but you do not indicate in a structure that protons are. May be you can indicate it in figure 1 adding colors that you can use when identify the protons in the different spectra.

RESPONSE: The labelling of protons in the whole work (figures and text) refers to the numbering in figure 1. The numbering in figure 1 was completed for the structures c,d and e.

In section 3.1 you speak about the influence of NMR solvents used. The explanation is clear, but to follow better, may be you can add the spectra in the supplementary materials.

RESPONSE: With Figure S1 (see Supplement Information) we had already made a selection of spectra. A clear and compact presentation of all spectra is difficult in our eyes because of the many different measurements (different solvents, samples and times). Therefore we decided to write this part in text form to give a better and more concise overview.